# A Cost-Effective and Rapid Manufacturing Approach for Electrochemical Transducers with Magnetic Beads for Biosensing

**DOI:** 10.3390/mi16030343

**Published:** 2025-03-17

**Authors:** Milica Govedarica, Ivana Milosevic, Vesna Jankovic, Radmila Mitrovic, Ivana Kundacina, Ivan Nastasijevic, Vasa Radonic

**Affiliations:** 1University of Novi Sad, Biosense Institute, Dr Zorana Djindjica 1, 21000 Novi Sad, Serbia; milica.govedarica@biosense.rs (M.G.); ivana.milosevic@biosense.rs (I.M.); ivana.kundacina@biosense.rs (I.K.); 2Institute of Meat Hygiene and Technology, Kacanskog 13, 11000 Belgrade, Serbia; vesna.jankovic@inmes.rs (V.J.); radmila.mitrovic@inmes.rs (R.M.); ivan.nastasijevic@inmes.rs (I.N.)

**Keywords:** electrochemical sensor, biosensors, gold leaf, pathogen detection, Salmonella, Listeria

## Abstract

Biosensors as advanced analytical tools have found various applications in food safety, healthcare, and environmental monitoring in rapid and specific detection of target analytes in small liquid samples. Up to now, planar electrochemical electrodes have shown the highest potential for biosensor applications due to their simple and compact construction and cost-effectiveness. Although a number of commercially available electrodes, manufactured from various materials on different substrates, can be found on the market, their high costs for single use and low reproducibility persist as major drawbacks. In this study, we present an innovative, cost-effective approach for the rapid fabrication of electrodes that combines lamination of 24-karat gold leaves with low-cost polyvinyl chloride adhesive sheets followed by laser ablation. Laser ablation enables the creation of electrodes with customizable geometries and patterns with microlevel resolutions. The developed electrodes are characterized by cyclic voltammetry and electrochemical impedance spectroscopy, scanning electronic microscopy, and 3D profiling. To demonstrate the manufacturing and biosensing potential, different geometries and shapes of electrodes were realized as the electrochemical transducing platform and applied for the realization of magnetic bead (MB)-labeled biosensors for quantitative detection of food-borne pathogens of *Salmonella typhimurium* (*S. typhimurium*) and *Listeria monocytogenes* (*L. monocytogenes*).

## 1. Introduction

Electrochemical sensors are powerful analytical tools known for their high sensitivity, rapid response, and broad applicability in healthcare, environmental monitoring, and food safety [1,2,3,4]. Nowadays, electrochemical sensors have found applications for rapid and specific detection of different targets such as ions, pathogens, heavy metals, or biomarkers. A crucial component of any electrochemical biosensor is the electrode, which transduces the chemical interaction between a biorecognition element (aptamers, nucleic acids, antibodies, enzymes, peptides, etc.) and the target into an electrical signal. Therefore, the choice of electrode material, geometry, and fabrication method plays a critical role in determining sensor performance, reproducibility, and cost [5].

In recent years, numerous fabrication technologies have been developed for the fabrication of electrochemical sensors. Electroplating is a widely used technique, offering precise control over electrode thickness, composition, and surface morphology [6,7]. This method enables the deposition of a wide range of conductive materials, including metals such as gold, platinum, and silver, onto various substrates [7]. One of the key advantages of this technology is the ability to create high-surface-area electrodes, which improve sensitivity and facilitate efficient electron transfer. Additionally, electroplating is a cost-effective and scalable process, making it suitable for large-scale sensor production. However, challenges such as non-uniform deposition, adhesion issues, and the need for strict control over plating parameters can affect sensor reproducibility. Moreover, impurities from plating baths may introduce defects, potentially influencing sensor stability and long-term performance. Despite these limitations, advancements in electroplating techniques, including pulse electrodeposition and template-assisted plating, are being explored to enhance the precision and functionality of electrochemical sensors.

Traditional methods such as physical vapor deposition (PVD) and chemical vapor deposition (CVD), in combination with photolithography or shadow masking, have been widely used to create thin-film electrodes on a variety of substrates [8,9,10,11]. PVD allows precise control of the thickness of the deposited film, providing excellent adhesion to the substrate [9], while CVD can be used to deposit a wide variety of materials, including metals, semiconductors, and insulators with a high resolution [10,11]. While these techniques produce highly precise and reproducible electrodes, they require expensive equipment, cleanroom facilities, and specific chemicals and reagents. Additionally, the thin films produced by PVD and CVD are realized on silicon, glass, or polymers as substrates that can be fragile, limiting their long-term durability and increasing the risk of damage during handling. PVD requires a high vacuum environment, which increases the complexity and cost of the process. Deposition occurs only on surfaces directly exposed to the vapor source, limiting the ability to coat complex geometries. On the other hand, CVD can be used for the realization of complex geometries and surfaces with high aspect ratios but requires chemically resistive substrates since it involves a chemical reaction. In addition, CVD produces toxic and corrosive precursor gases, which demand rigorous safety measures and chemical waste management.

Screen printing is a widely used technique for the mass production of planar electrochemical sensors due to its scalability, cost-effectiveness, and compatibility with various substrates, including ceramics, glass, and polymers [12,13]. It enables rapid fabrication of patterned electrode structures using conductive inks or pastes through a screen mask. Additionally, its adaptability allows for the development of flexible and wearable sensors by printing on polymer or textile materials [14]. Moreover, the integration of conductive nanomaterials, such as graphene, carbon nanotubes, and metallic nanoparticles, has been widely used in enhancing conductivity and sensor sensitivity [15]. However, reproducibility remains a major challenge, as screen imperfections, such as inconsistencies in mesh size and thickness, can lead to variations in electrode patterns, affecting electrochemical performance. In addition, the inks used in screen printing often contain organic binders that, while necessary for printability and adhesion, can introduce impurities that interfere with biorecognition layers, potentially reducing sensor performance. Therefore, post-printing treatments, such as curing or sintering, are often used to mitigate those problems. While screen printing remains the dominant method for fabricating disposable electrochemical sensors, and different screen-printed electrodes are available on the market, further optimization is required to obtain high-precision electrodes, and their price is quite high for single use.

Inkjet printing has emerged as a promising technology for the development of electrochemical sensors, enabling the precise fabrication of patterned electrodes on various flexible substrates [16,17,18]. This method eliminates the need for masks or complex lithography, thus reducing production costs and complexity. One significant advantage of inkjet printing is its ability to create intricate patterns with microscale resolution, making it suitable for applications such as bio-sensing and diagnostics. Furthermore, inkjet printing allows for the deposition of a wide range of conductive inks, which can be tailored to enhance the performance of the sensors [17]. However, technology does face some limitations, notably the requirement for post-printing sintering of inks, which can restrict its use with heat-sensitive materials. Additionally, the cost of high-quality conductive inks and specialized inkjet printers hinders broader adoption in the market. Despite these challenges, recent advancements in inkjets demonstrate the application of different nanoparticles for the realization of electrochemical electrodes applicable in food safety and healthcare to detect different biological targets, such as proteins, pathogens and toxins [18]. Moreover, innovations like 3D inkjet printing can further expand capabilities, allowing for complex 3D electrode architectures [19].

Additive manufacturing or 3D printing offers significant advantages in the fabrication of miniaturized electrochemical sensor electrodes [20,21,22]. This technology allows precise control over electrode geometry, enabling the creation of complex microstructures and flexible, wearable sensors. It supports a wide range of materials, including conductive polymers and metals, enhancing the performance and sensitivity of electrochemical sensors. Moreover, 3D printing can integrate multiple sensing components, reducing manufacturing costs and streamlining the fabrication process. However, challenges remain, such as limited resolution and material conductivity, which may impact sensor precision and performance. Additionally, post-processing treatments like sintering are often required to improve material properties. Despite these limitations, 3D printing holds promise for cost-effective, scalable production of disposable biosensors for various applications [22].

The selection of electrode material also significantly impacts the performance of electrochemical sensors. Gold has been identified as one of the most suitable materials for electrochemical biosensors due to its excellent electrical conductivity, chemical stability, and biocompatibility. Gold’s affinity for binding molecules modified with thiol groups makes it an ideal substrate for immobilizing a variety of biorecognition elements. However, conventional gold electrode fabrication techniques, such as CVD and PVD, are costly and environmentally challenging, leading to the exploration of alternative fabrication methods.

Electrochemical sensors incorporating magnetic beads (MBs) have gained significant attention due to their ability to enhance sensitivity and selectivity in bioanalytical applications [23,24]. The use of MBs facilitates efficient target capture, preconcentration, and signal amplification, particularly in enzymatic and immunosensing platforms [24]. Magnetic beads also play a crucial role in preconcentration and separation strategies, enabling the accumulation of low-abundance analytes and improving detection limits [25,26]. Recent advances include the integration of nanomaterials such as gold nanoparticles and carbon nanostructures, further enhancing electron transfer and analytical performance [27]. Additionally, MB-based electrochemical biosensors have demonstrated excellent performance in detecting biomolecules like DNA, proteins, and small metabolites with high specificity [28]. These developments position MB-enhanced electrochemical biosensing as a promising tool for medical diagnostics, environmental monitoring, and food safety applications.

In this paper, a cost-effective and rapid approach for a label-free MB biosensor was proposed for application in pathogen detection. Electrochemical transducers based on gold leaf electrodes (GLEs) were fabricated combining gold leaf lamination with polyvinyl chloride (PVC) adhesive sheets and laser ablation to pattern the electrode geometry. This technique allows the rapid production of highly conductive electrodes with large surface areas, providing a suitable platform for the immobilization of aptamers, antibodies, and other biomolecules [29,30,31]. To demonstrate the potential of the proposed technology, different geometries and shapes of electrodes were realized and characterized using galvanostatic and impedimetric methods, scanning electronic microscopy, and 3D profiling. Two electrochemical transducing platforms were used for the realization of MB-labeled biosensors for quantitative detection of pathogens important for food safety and public health, such as *S. typhimurium* and *L. monocytogenes*.

## 2. Materials and Methods

### 2.1. Chemicals

All solutions were prepared in MiliQ water. Potassium ferricyanide (K_3_[Fe(CN)_6_]), potassium ferrocyanide (K_4_[Fe(CN)_6_]), and sulfuric acid (H_2_SO_4_) were all purchased from Sigma-Aldrich (Saint Louis, MO, USA) and used as received. A stock solution of PBS (10×) was purchased from Fisher Bioreagents (Loughborough, UK), pH 7.4. Magnetic bead APD50: Pathatrix™ Dual (*Listeria*/*Salmonella* spp.) Kit Phatarix APD 50 was purchased from Thermo Fisher Scientific Inc. (Waltham, MA, USA). As a redox probe, a 10 mM ferri/ferrocyanide redox couple in PBS was used.

### 2.2. Materials for Electrode Fabrication and Characterization

For electrode fabrication, dry lubricant polytetrafluoroethylene (PTFE) spray was purchased from Wurth (Belgrade, Serbia), PVC foils “ImageLast A4 125 μm Laminating Pouch” were purchased from Fellowes Brands (Marki, Poland) and gold leaves were purchased from Noris Blattgoldfabrik (Schwabach, Germany).

### 2.3. Fabrication of Gold Leaf Electrodes

The fabrication process of GLEs is shown in Figure 1. To fabricate GLEs, two gold leaf foils (with size of 80 mm × 80 mm) were placed onto four-layer PVC sticker foil (4 × 125 μm thickness), followed by light pressing at 135 °C using Laminator Fellowes 5709001 (Proteus, Shanghai, China). PTFE spray was applied before the gold deposition to make the surface of the PVC sticker hydrophobic and to prevent liquid samples from spreading on the electrode surface during measurements. Layouts of custom-made GLEs were patterned by laser ablation of gold using a Nd:YAG laser Power Line D-100 (Rofin-Sinar, Hamburg, Germany) in hatch mode and at 26.2 A current, 65 kHz frequency and 500 mm/s speed. Different configurations and geometries of GLE with two, three and five electrodes were realized including interdigitated electrodes, circular three-electrode systems, circular three-electrode systems with different working electrode patterns (rhomboidal, snowflake and circular), and multiplex GLE platform composed of four working electrodes and one counter electrode, as shown in Figure 2. Layouts of the fabricated prototypes of GLEs are shown in Figure 3. All GLEs were cleaned by cyclic voltammetry (CV) in the presence of 0.5 M sulfuric acid for ten cycles in a potential window between −0.2 and 1.3 V vs. Au reference electrode at a scan rate of 0.5 V/s. The electrodes were then rinsed with MiliQ water and dried.

### 2.4. Bacteria Cultivation, Enumeration, and Sample Preparation

*S. typhimurium* ATCC 13311 (Microbiologics, St. Cloud, MN, USA) was retrieved from its commercial stock culture 48 h before testing to ensure an adequate volume of bacterial suspension needed for the necessary serial dilutions and concentrations (10^1^–10^5^ CFU/mL). The preparation process involved dissolving the nutritional broth (ONE Broth-Salmonella Base, Oxoid, UK) in distilled water, sterilizing it via autoclaving at 121 °C for 15 min, and allowing it to cool to room temperature, forming a liquid medium suitable for Salmonella proliferation. For incubation, 100 μL of the prepared suspension was transferred into sterile culture tubes for each strain and incubated at 37 °C for 18–24 h, facilitating bacterial growth and providing a fresh culture for subsequent testing.

For *L. monocytogenes* ATCC 19115 (Microbiologics, St. Cloud, MN, USA), recovery from the freeze-dried commercial stock culture involved rehydrating the bacterial pellet, inoculating 100 μL onto a Brain Heart Infusion (BHI) agar plate, and incubating it overnight at 37 °C for 18 h. A single, freshly isolated colony was then transferred to BHI broth in a culture tube and incubated again under the same conditions to ensure active and fresh bacteria. The resulting culture underwent serial dilutions (10^1^–10^5^ CFU/mL) for use in testing.

The *Escherichia coli* (*E. coli*) strain ATCC 25922 was cultivated by inoculating 100 μL of the sample onto Tryptic Soy Agar (TSA) plates, followed by overnight incubation at 37 °C for 18–24 h. A single bacterial colony was subsequently transferred to BHI broth, with the turbidity of the culture adjusted to a 0.5 McFarland standard. Serial dilutions (10^1^–10^5^ CFU/mL) were prepared in phosphate-buffered saline (PBS). To validate bacterial concentrations, duplicate quantitative counts were performed by plating 0.1 mL of the 10^2^ and 10^5^ CFU/mL dilutions onto TSA plates.

The recovery of *Staphylococcus aureus* (*S. aureus*) ATCC 25923 (Microbiologics, St. Cloud, MN, USA) involved rehydrating the freeze-dried bacterial pellet and transferring 100 μL of the suspension into BHI broth. The culture was incubated overnight at 37 °C for 18–24 h. A single bacterial colony was then transferred to TSA plates using sterile culture tubes and subjected to an additional overnight incubation under the same conditions to obtain a fresh culture. The same range of serial dilutions (10^1^–10^5^ CFU/mL) were subsequently prepared for use in the study.

To achieve uniform bacterial distribution within the suspensions, a vortex mixer (Stuart™ Scientific SA8, Merck, Darmstadt, Germany) was employed following the final overnight incubation. Serial dilutions were conducted by transferring a portion of the bacterial suspension from one tube to the next, using fresh sterile pipette tips for each transfer to prevent cross-contamination.

For evaluating the sensitivity and specificity of the electrochemical sensors, test samples including single-culture preparations from commercial stock cultures were spiked with individual bacterial cultures (*S. typhimurium*, *L. monocytogenes*, *E. coli* and *S. aureus*).

The sample was first incubated with 5 µL with MBs for 2 min and vortexed. After incubation, magnetic separation was performed to isolate the pathogen, followed by the removal of the sample. The captured beads were then transferred onto the GLE surface, where 80 µL of redox probe was added for measurements.

### 2.5. Electrochemical Measurements

Electrochemical measurements were performed with a Potentiostat/Galvanostat/Impedance Analyzer-PalmSens 4 (PalmSens BV, Houten, The Netherlands), connected to a personal computer equipped with PSTrace 5.8 software. CV measurements with a 10 mM ferro/ferricyanide redox probe in PBS were performed in the potential range from −0.3 V to 0.3 V, with a scan rate of 0.5 V/s. A third Au electrode was used as the reference electrode in CV measurements. Impedimetric measurements were carried out with the ferro/ferricyanide redox probe over a frequency range from 1 Hz to 100 kHz, with the potential amplitude of 10 mV. The direct potential during measurements was set to 0 and all the measurements were performed versus an open circuit potential.

EIS was modeled with equivalent circuit models enabling interpretation of the electrical properties of the electrical double layer. The open-source modeling program EIS Spectrum Analyzer [32] was used to model the impedance spectra with an equivalent Randles circuit. The program is available at https://www.abc.chemistry.bsu.by/vi/analyser/, 20 February 2025.

### 2.6. Scanning Electron Microscopy (SEM) and Profiler Analysis

The sample preparation and morphological analysis were performed using an Ap-reo 2 C Scanning Electron Microscope (SEM, Thermo Fisher Scientific, Waltham, MA, USA). Electrodes were fixed on aluminum stubs using carbon adhesive tape, with connections made at the bottom to keep them stable during imaging. Samples underwent a dehydration protocol involving fixation in glutaraldehyde, sequential ethanol dehydration, and chemical drying with Hexamethyldisilane (HMDS) to stabilize them on the electrode surface. Imaging was performed in high vacuum mode using secondary electrons, with the SEM operating at a beam energy of 500 eV and a beam deceleration of 4 kV. The T2 detector was used, with the working distance adjusted between 2 mm and 10 mm depending on the sample requirements.

The Profiler Huwitz HRM-300 (Seoul, Republic of Korea) series microscope with Panasis imaging software ver. 2.5.39 was used for making 2D and 3D profiles of the GLE surface. Leica Ivesta 3 Greenough Stereo Microscopes was used for GLE imaging.

## 3. Results and Discussion

The analysis first starts with basic characterizations of all fabricated electrodes. For standard circular three-electrode GLEs, shown in Figure 2a, we conduct stability and reproducibility tests, which are crucial for ensuring the reliability of custom-made electrodes. The CV diagram shown in Figure 4a shows a single oxidation and reduction peak at ±0.1 V corresponding to the electrochemical reduction/oxidation of the redox couple on the gold layer in the forward scan and backward scan. Such a CV curve is typical for bare gold with the ferri/ferrocyanide redox couple. In addition, multiple CV scans show that the ferri/ferrocyanide redox reaction displayed well-defined oxidation and reduction peaks since their intensities had constant values after 15 successive scans, indicating an excellent stability of the GLE. EIS was employed to evaluate the reproducibility of the electrodes in the ferri/ferrocyanide redox couple solution in PBS (Figure 4b). The impedance spectra featured a semicircular region, whose diameter corresponds to the electron transfer resistance, followed by a linear region indicative of a diffusion-controlled process. The electrodes demonstrated good reproducibility, as five randomly fabricated devices exhibited highly consistent responses with minimal variability. The GLEs were further characterized by CV using different scan rates. As shown in Figure 4c, the current increased proportionally with the scan rate varying from 0.1 to 1 V/s. Plotting the anodic and cathodic peaks against the square root of the scan rate, as shown in Figure 4d, revealed a linear relationship, indicating a diffusion-controlled redox process on the realized GLE.

Since the geometry of electrodes and their surface area can influence the response, we investigated the different configurations of circular electrodes with different deformation patterns, namely rhomboidal (Figure 2b), with 9 circular openings (Figure 2c), and with 13 circular openings distributed in a snowflake configuration (Figure 2d). Comparisons of CV and EIS responses for these configurations are presented in Figure 3e and Figure 3f, respectively. Depending on geometry, impedance responses are significantly changed due to changes in the electroactive surface of the electrode. The electroactive surface was calculated according to the Randles–Sevcik equation for the reversible electrode process:(1)$$Ip=2.69\;\cdot 10^{5}\cdot A\cdot C\cdot n^{3/2}\cdot D^{1/2}\cdot v^{1/2}$$
where *A* is the electroactive area in cm^2^, *Ip* is an anodic current peak in *A*, *D* is the diffusion coefficient (6.1·10^6^ cm^2^/s in the case of the used redox solution), *n* is the number of electrons transferred in half-reaction (1 for redox solution), *ν* is scan rate (where 0.5 V/s was used) and *C* is the concentration of the redox probe in mol/cm^3^. Based on that, the electroactive surfaces of all compared geometries were calculated. For circular GLE with rhomboidal deformation, the calculated electroactive surface is 19.89 mm^2^, for circular electrode with 9 circular openings as the deformation, it is 19.61 mm^2^, for circular electrode with snowflake deformation, it is 15.01 mm^2^, while the standard circular electrode has 22.72 mm^2^. The results show that the electrode pattern significantly affects the electroactive surface area. The more gold surface is removed, the more the electroactive surface decreases compared to the standard circular electrode, resulting in increased impedance.

Two additional geometries are proposed, two-electrode GLEs with interdigitated electrodes, shown in Figure 2e, suitable for impedance measurements, and a multiplex GLE comprising four working electrodes, and one big counter electrode which surrounds four working electrodes (Figure 2f), ideal for multiple detections. The EIS response of interdigitated electrodes for 5 randomly fabricated electrodes is shown in Figure 4g, confirming good impedance reproducibility. EIS results for multiplex GLEs on four working electrodes, shown in Figure 4h, confirm the repeatable response on all four working electrodes. The proposed results emphasize the strong potential of the developed technology, demonstrating the stable and reliable performance of GLEs. By exploring various electrode geometries and configurations, we can develop low-cost biosensors that enhance sensitivity and selectivity, ultimately facilitating the more effective realization of biosensors for the detection of various targets across a range of applications.

To further demonstrate the advantage of the proposed technology in achieving electrodes with high resolution, enlarged sections of the fabricated electrodes are shown in Figure 5a. It is evident that the micron resolution can easily be reached since it is determined by the laser beam width of 30 μm. Unfortunately, this is an advantage that very few technologies can achieve without utilizing the lithography process and expensive micro- and nanofabrication facilities. Two-dimensional and three-dimensional profiles of the interdigitated GLEs are presented in Figure 5b and illustrate the roughness of the gold surface. The surface of the GLE showed regular undulating features with micrometer-scale hills and valleys. This is another advantage of the proposed technology, as the surface itself is sufficiently rough, leading to enhanced sensitivity. Consequently, the addition of various nanomaterials or nanoparticles will have an insufficient impact on further improving the GLE performances.

To demonstrate the potential of GLEs in the realization of the biosensors, MB-labeled biosensors for quantitative detection of two food-borne pathogens have been developed and tested, namely biosensors based on GLE with rhomboidal deformation for *S. typhimurium* detection and interdigitated GLE biosensors for *L. monocytogenes* detection. The developed GLEs were tested on increasing concentrations of selected pathogens in PBS. For both pathogens, a concentration of 10^1^–10^5^ CFU/mL was prepared and spiked in PBS. The sample was first incubated with MBs and vortexed. After incubation, magnetic separation was performed, followed by transferring MBs onto the GLEs’ surface, where a redox probe was added for measurements. Each pathogen concentration was tested with three measurements. During the testing, all pathogen concentrations were tested on the same GLE.

First, we tested the developed GLE with rhomboidal deformation on *S. typhimurium*. Nyquist plots of the impedance spectra for different concentrations of *S. typhimurium* are reported in Figure 6a. The impedimetric measurements were fitted using an equivalent circuit of the Randles cell shown in the inset of Figure 6b and the corresponding electric circuit elements were calculated. In brief, the circuit includes four elements: the ohmic resistance of electrolyte solution (*R*), where the *R_ct_C* circuit represents the resistance to charge transfer at the electrode/electrolyte interface and the Warburg impedance (*W*), which is related to the low-frequency part of the spectrum, where the diffusion mechanism controls the measured impedance and charge transfer resistance *R_ct_*. Of these elements, the major change was an increase in the charge transfer resistance *R_ct_*, which is related to the mass transfer phenomenon and/or the dielectric or conductive properties of the captured bacterial cells. In the proposed biosensors, *R_ct_* changes when the MBs or MBs with bacteria are present at the electrode surface, affecting the electron transfer kinetics. A decrease in *R_ct_* is visible when just MBs are present on the sensor surface due to improved surface conductivity, while an increase in *R_ct_* indicates change due to pathogen binding. Therefore, the charge transfer resistance was used to quantify bacteria. The relative change of the signal on the electrode was calculated as the ratio of the signal on the electrode with bacteria relative to the electrode with only MBs as:(2)$$\delta =\frac{R_{ct(sample)}–R_{ct(MBs)}}{R_{ct(MBs)}}\times 100\%$$

The relative change of the signal is shown in Figure 6b, where a good linear dependence can be observed in the range of 10^1^–10^5^ CFU/mL and regression equation y = 269.04x + 72.872, with R^2^ of 0.9808. The experimental LoD was 10 CFU/mL, while the calculated LoD was 1.74 CFU/mL in PBS. The LoD was calculated as 3*S*/d, where *S* represents the standard deviation of a blank test measured with three biosensors, and *d* denotes the sensitivity derived from the slope of the linear curve. Furthermore, the selectivity was studied on Gram-negative bacteria *E. coli* and Gram-positive bacteria *S. aureus* as non-specific bacterial targets for the capturing MBs (Figure 6c). The control non-specific bacterial cells were introduced on the electrode surface using the same experimental protocol. The relative change in *δ* indicated that no significant response was detected for both control bacteria in the concentration range up to 10^5^ CFU/mL compared to the signal increase observed with 10^1^ CFU/mL of *S. typhimurium*.

Interdigitated GLE was used for the detection of *L. monocytogenes* and tested on increasing concentrations of selected pathogens in PBS. Nyquist plots of the impedance spectra for different concentrations of *L. monocytogenes* are reported in Figure 7a. The calibration curves presented in Figure 7b indicate the linear relative change of the signal within the observed range of 10^1^–10^5^ CFU/mL, where the regression equation is y = 75.354x + 132.12, with R^2^ of 0.8861. It should be noted that after 4 CFU/mL, the sensor slowly enters saturation, and therefore, the last point has the largest deviation from the linear curve. If we exclude the last point, R^2^ is 0.9424, demonstrating satisfactory linearity. Hence, an additional interpolation with two linear ranges was introduced in Figure 7c. The experimental LoD was 10 CFU/mL, while the calculated LoD was 2.65 CFU/mL. The selectivity response for *E. coli* and *S. aureus* as non-specific bacterial targets is shown in Figure 7c. The relative change in δ showed no significant response for both control bacteria across concentrations up to 10^5^ CFU/mL, in contrast to the noticeable signal increase observed at 10^1^ CFU/mL of *L. monocytogenes*.

To confirm efficiency and binding ability of the GLE surface, we performed SEM visualization. GLE surfaces with MBs before and after measurement are shown in Figure 8. It can be observed that the magnetic particles are non-uniform in size, varying from 100 µm to 1 mm with a tendency to agglomerate on the surface of the sensor. Magnified SEM images of GLE showed a random distribution of dark spots corresponding to bonded bacteria, showing that MBs specifically bind both targeted bacteria.

The laser ablation method shown in this study provides a cost-effective and rapid alternative to expensive micro- and nanofabrication techniques like photolithography while maintaining high resolution of electrode fabrication. Traditional fabrication methods often require high vacuum environments and cleanroom facilities, which increase their cost and complexity [33]. Additionally, commercial planar electrodes often involve multi-step processes including photolithography, etching, screen printing, or inkjet printing, all of which require time-consuming fabrication processes, cleanroom environments or multiple processing steps [1]. These methods significantly increase production time, often taking several hours per batch, and raise manufacturing costs due to the need for specialized materials and equipment [34]. In contrast, the described manufacturing approach eliminates the necessity for support layers and postprocessing. Unlike conventional methods, which struggle with complex electrode geometries, laser ablation enables flexible patterning of various shapes such as rhomboidal, circular, and interdigitated electrodes with minimal spacing, without additional masks or subprocesses overperforming limitations of mostly commercially available technologies. With a production time of under a minute per electrode and the capability to produce multiple copies simultaneously, it outperforms traditional methods in terms of efficiency. The affordability of approximately EUR 0.1 per copy further highlights its economic advantage compared to conventional methods. Therefore, the proposed technique with micron-scale precision is promising for large-scale production due to its efficiency and lower costs. Furthermore, in terms of surface roughness, the GLEs fabricated exhibited rippling micrometer-scale features, further enhancing GLE sensitivity, which is not achievable with other technologies such as ink-jet printing, photolithography, or sputtering, in which electrodes are characterized by relatively flat surfaces. Moreover, the electrochemical results obtained with the GLEs in this study, in terms of sensitivity and stability, demonstrate a good and repeatable performance that is comparable to, if not better than, electrodes fabricated by more traditional methods. Demonstrated applications in biosensor development show a potential to significantly impact biosensing technologies, offering a low-cost electrochemical platform for healthcare, agriculture, food safety, and environmental monitoring. While commercial electrodes remain widely used due to established manufacturing processes and regulatory approvals, the proposed method presents a disruptive potential for future sensor development since industries seek more cost-efficient and scalable solutions. Table 1 compares the proposed gold leaf MB-labeled biosensors with similar biosensors previously reported in the literature. The proposed solutions show comparable or better results in terms of LoD compared with advanced biosensors recently proposed in the literature. Although they exhibit a slightly lower performance in detecting higher concentrations than most biosensors, they can detect low pathogen concentrations required in the management of food safety systems and consumer protection strategies.

The biosensor proposed in [35] is based on the molecularly imprinted polymer with specific binding cavities functionalized on screen-printed electrodes, while the biosensor proposed in [36] presents a microfluidic antibody-based biosensor with quantum dots (QDs) as fluorescent probes for sensor readout and manganese dioxide nanoflowers and as QD nanocarriers for signal amplification. Disposable electrochemical immunoassay uses lysine-magnetic nanoparticles modified with anti-*E. coli* antibody and glutaraldehyde for their covalent immobilization for quantitative bacteria detection [37]. Magnetic Au nanoparticles are used for preconcentration followed by surface-enhanced Raman spectroscopy measurements to detect *L. monocytogenes* in [38]. The nanoenzyme-based electrochemical sensor was developed in [39] to detect *L. monocytogenes* based on aptamer-regulated platinum nanoparticles/hollow carbon spheres in [40], showing a similar detection limit as our proposed solution. Two *L. monocytogenes* biosensors with carbon nanotubes were proposed. First, an impedimetric biosensing platform was developed by immobilizing bacteriophages as bioreceptors onto quarternized polyethylenimine modified carbon nanotubes [39] and another gold-plated wire biosensors functionalized using single-walled carbon nanotubes, polyethyleneimine (PEI) and antibodies [41]. Despite the complexity of existing sensor designs, our solution stands out for its simplicity and achieved high sensitivity without the need for additional signal-enhancing materials or layers. The proposed technology offers cost-effective, rapid fabrication while delivering an equal or even better limit of detection.

## 4. Conclusions

In this paper, a novel, cost-effective, and rapid approach for fabricating an electrochemical transducing platform based on GLEs with magnetic beads for biosensing applications was proposed. The various electrode geometries, including rhomboidal GLE and interdigitated configurations, exhibited excellent stability, reproducibility, and sensitivity, with minimal signal variation across multiple electrode series. The developed biosensors were successfully applied for the label-free detection of *S. typhimurium* and *L. monocytogenes*, demonstrating their potential for quantitative pathogen detection in food safety applications. Impedance spectroscopy analysis of the biosensors indicated a good linear correlation between the charge transfer resistance and pathogen concentration. The proposed MB-labeled biosensor platform showed high specificity, as no significant response was detected for non-specific bacteria *E. coli* and *S. aureus*, confirming excellent sensor selectivity. The experimental limit of detection (LoD) for both pathogens was 10 CFU/mL, demonstrating the high sensitivity of the proposed biosensors. Therefore, the developed GLE shows a good potential for the realization of sensitive biosensors since the proposed technology presents a promising, scalable solution for rapid and cost-effective detection, with potential applications in healthcare, food safety and environmental monitoring.

## Figures and Tables

**Figure 1 micromachines-16-00343-f001:**
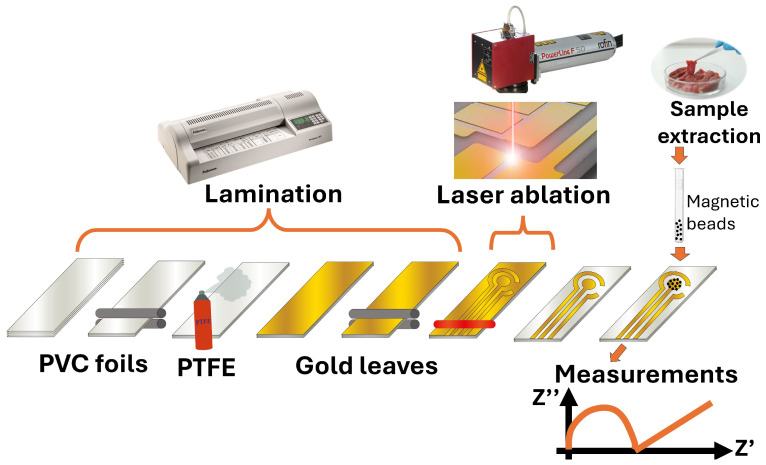
GLE fabrication and measurement processes. The fabrication process encompasses two steps: lamination and laser ablation. Measurement with magnetic beads involves capturing target analytes on functionalized beads, magnetically concentrating them at the electrode surface, and measuring the electrochemical signal generated by redox reactions.

**Figure 2 micromachines-16-00343-f002:**
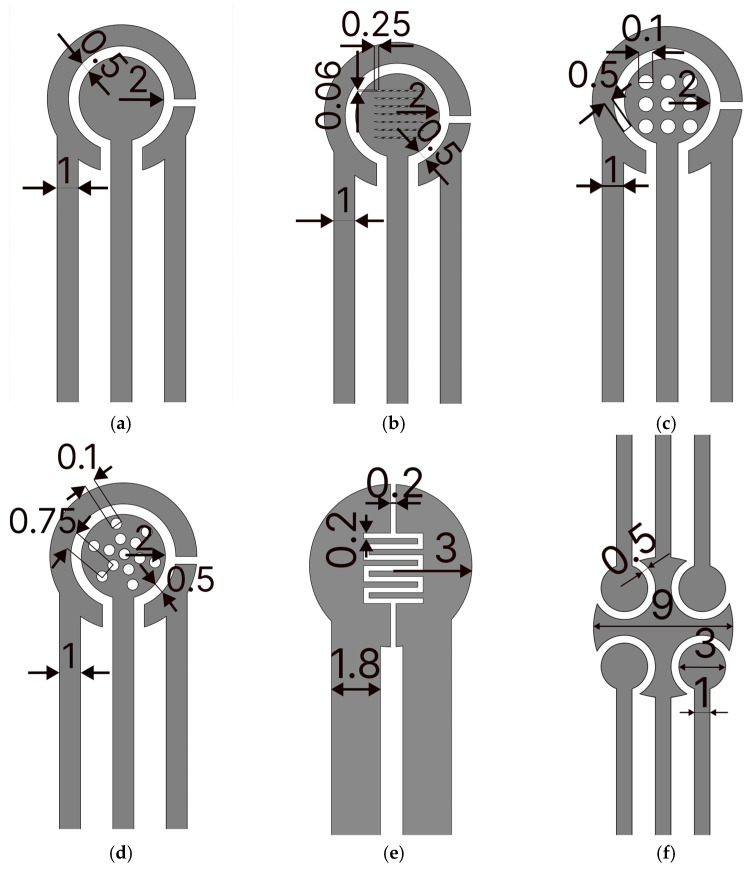
Layouts of fabricated designs: (**a**) circular GLE, (**b**) circular GLE with rhomboidal deformations, (**c**) circular GLE with nine circular deformations, (**d**) circular GLE with snowflake deformations, (**e**) interdigitated electrode, (**f**) multiplex GLE. All dimensions are in mm.

**Figure 3 micromachines-16-00343-f003:**
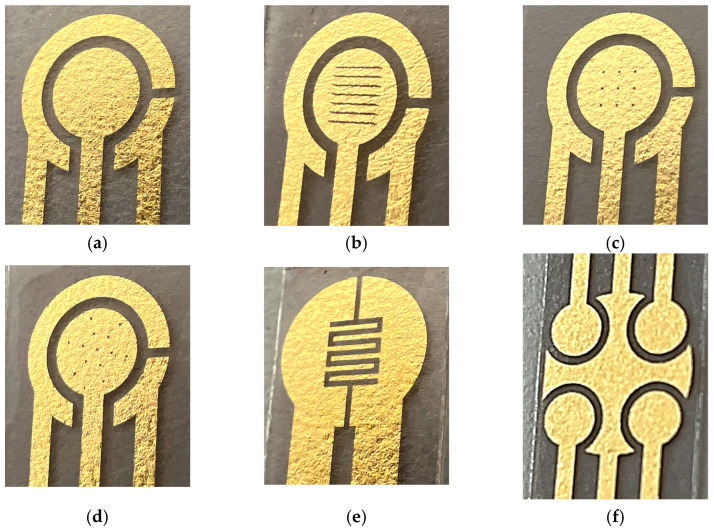
Fabricated prototypes: (**a**) circular GLE, (**b**) circular GLE with rhomboidal deformations, (**c**) circular GLE with nine circular deformations, (**d**) circular GLE with snowflake deformations, (**e**) interdigitated electrode, (**f**) multiplex GLE.

**Figure 4 micromachines-16-00343-f004:**
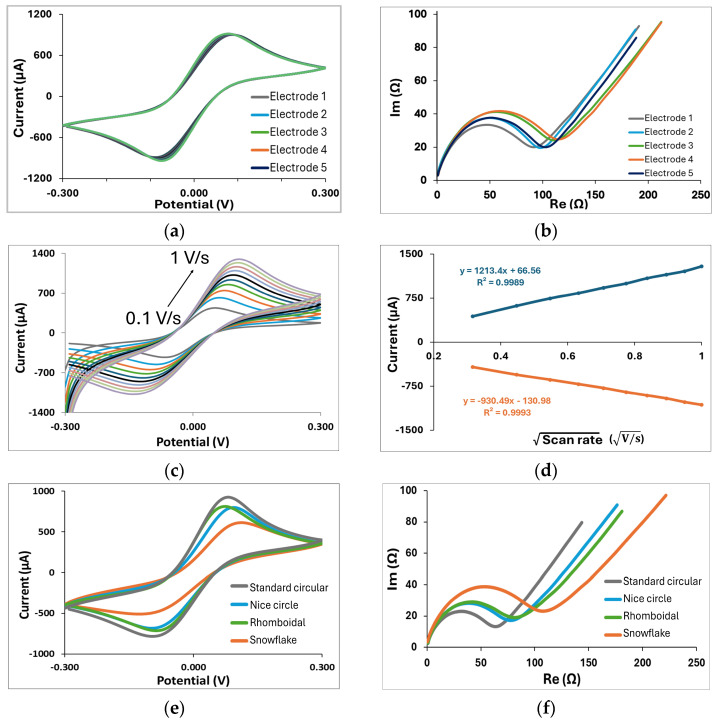
Standard characterization of all fabricated electrodes: (**a**) CV response and stability CV of circular GLE recorded using 15 successive time cycles on the same electrode, (**b**) EIS reproducibility test for 5 randomly selected circular GLEs, (**c**) CV of circular GLE at different scan rates from 0.1 to 1 V/s, (**d**) the dependence of anionic and cationic peak intensity on the square root of scan rate for circular GLE, (**e**) CV responses of circular GLEs with different deformation patterns, (**f**) EIS responses of circular GLEs with different deformation patterns, (**g**) EIS reproducibility test for 5 randomly selected interdigitated GLEs, (**h**) EIS response for multiplex GLEs on four working electrodes.

**Figure 5 micromachines-16-00343-f005:**
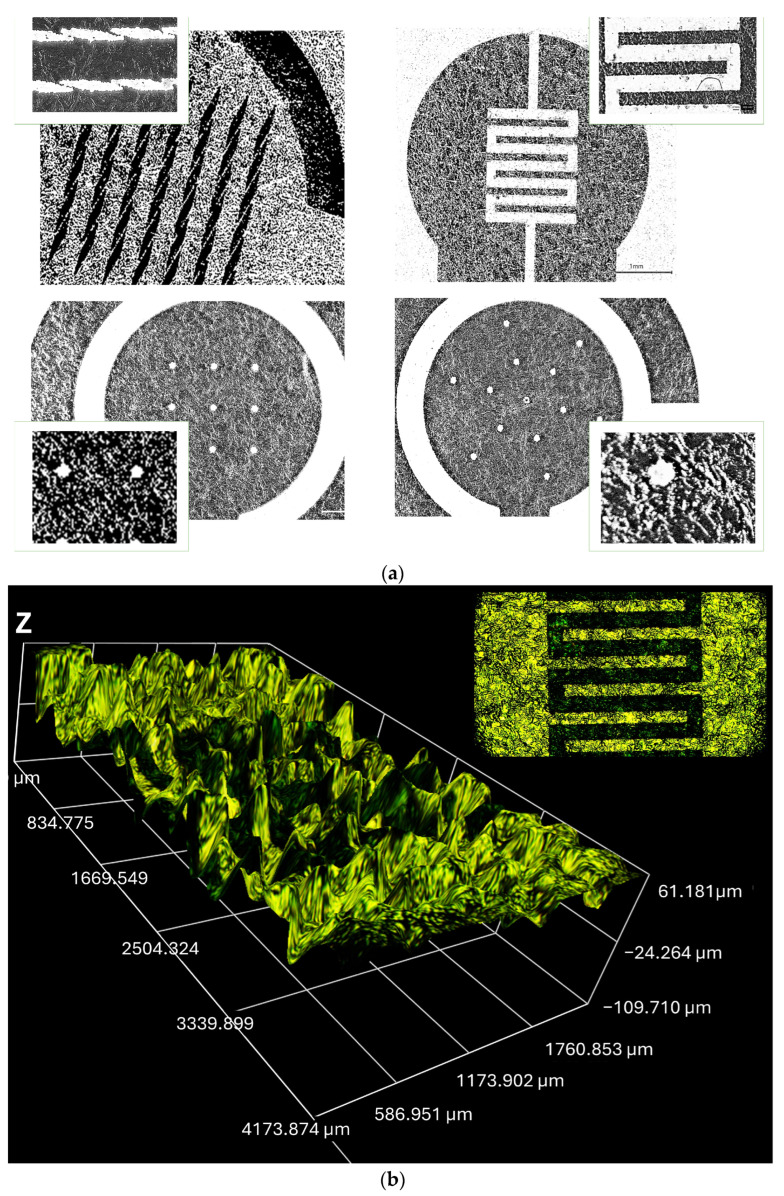
Image and profile of fabricated electrodes: (**a**) image of the fabricated GLEs with enlarged sections with deformations, (**b**) 2D and 3D profiles of interdigitated GLE.

**Figure 6 micromachines-16-00343-f006:**
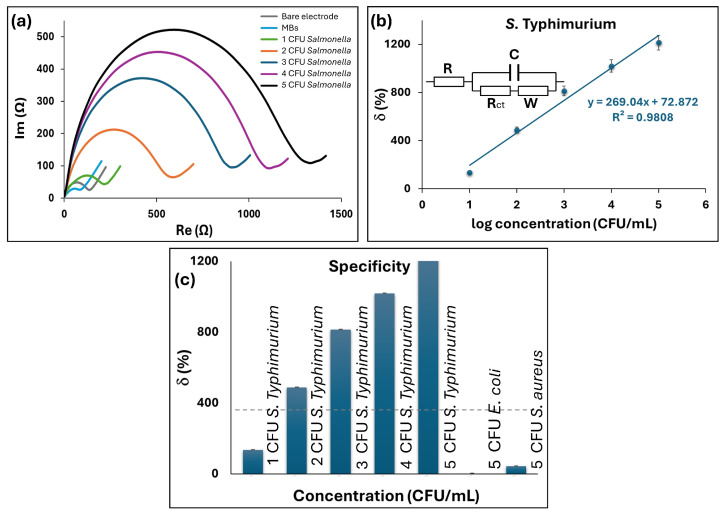
Characterization of *S. typhimurium* MB-labeled biosensors on GLEs with rhomboidal deformation: (**a**) Nyquist plots for different concentrations of *S. typhimurium* in PBS, (**b**) calibration curve as a function of the different concentrations of *S. typhimurium*, (**c**) specificity test for different pathogens *E. coli* and *S. aureus*.

**Figure 7 micromachines-16-00343-f007:**
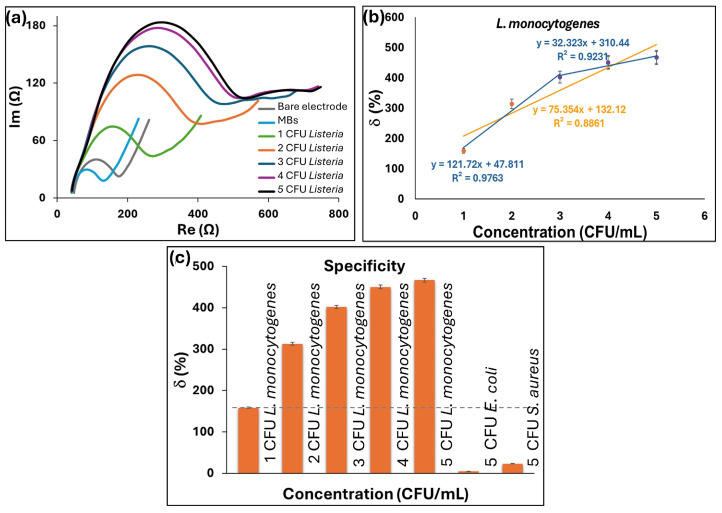
Characterization of *L. monocytogenes* MB-labeled biosensors on interdigitated GLE: (**a**) Nyquist plots for different concentrations of *L. monocytogenes* in PBS, (**b**) calibration curve as a function of the different concentrations of *L. monocytogenes*, linear interpolation and interpolation with two linear ranges, (**c**) specificity test for different pathogens *E. coli* and *S. aureus*.

**Figure 8 micromachines-16-00343-f008:**
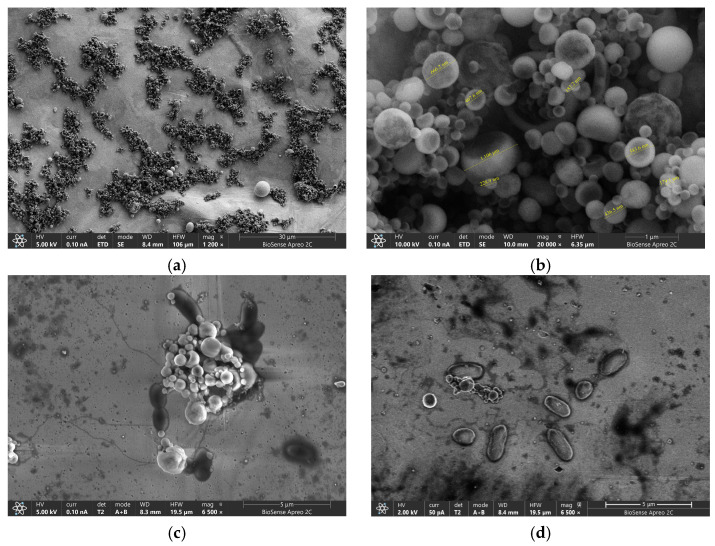
SEM images of the GLE surface: (**a**) MBs on electrode surface, (**b**) magnified MBs showing variation in bed size, (**c**) image of the biosensor electrode with *S. typhimurium*, (**d**) image of the biosensor electrode with *L. monocytogenes*.

**Table 1 micromachines-16-00343-t001:** Comparison of the proposed gold leaf MB-labeled biosensors with biosensors previously reported in literature.

Biosensor Type	Target	Detection Range (CFU/mL)	LoD (CFU/mL)	Reference
Electrochemical	*S. typhimurium*	10^1^–10^5^	10	[35]
Microfluidic	*S. typhimurium*	10^2^–10^7^	43	[36]
Electrochemical	*E. coli*	266–10^6^	80	[37]
Surface-Enhanced Raman Scattering	*L. monocytogenes*	10^2^–10^6^	12	[38]
Electrochemical	*L. monocytogenes*	10^1^–10^4^	8.4	[39]
Electrochemical	*L. monocytogenes*	10^1^–10^9^	2	[40]
Electrochemical	*L. monocytogenes*	10^3^–10^8^	10^3^	[41]
This work	*S. typhimurium* *L. monocytogenes*	10^1^–10^5^	1.74 2.65	

## Data Availability

The datasets presented in this article are not readily available because the data are part of an ongoing project MicroLAbAptaSense. Requests to access the datasets should be directed to vasarad@biosense.rs.

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
