# Peer review of "A Cost-Effective and Rapid Manufacturing Approach for Electrochemical Transducers with Magnetic Beads for Biosensing"

_micromachines, 2025, doi:10.3390/mi16030343_

Round 1
Reviewer 1 Report
Comments and Suggestions for Authors
The manuscript describes the development of the approach for the rapid fabrication of electrodes with gold and PVC followed by laser ablation. This approach allowed to produce electrodes with different geometry, shape and area. The electrodes were thoroughly tested with a range of electrochemical, microscopic and profile imaging methods. The approach presented is interesting and perspective. Two types of the gold electrodes were used for the fabrication of magnetic beads labeled biosensors for food pathogen detection. To date, devices for food safety control are highly demanded. The references are relevant to the manuscript topic and their quantity is sufficient. This manuscript can be published in Micromachines journal after several minor revisions:
1) What was the pH value of PBS used?
2) Line 247: HMDS - add the abbreviation description to the manuscript text.
3) Lines 28, 151: replace “labeling “with “labeled”, Lines 271, 292-293: replace “anionic” and “cationic” with “anodic” and “cathodic”, Lines 353-354: replace “electronic” with “electric”, Line 220: replace “spiked” with “were spiked”.
4) There is a poor phrase in the manuscript: Line 441: “The different electrode geometries, including rhomboidal GLE and interdigitated configurations, were shown demonstrating excellent stability, reproducibility, and sensitivity, with minimal signal variation across series of electrodes.”,
5) Figures 4e and 4f are the same. Please insert the appropriate Figure.
6) Please check the captions to Figures 6c and 7c. Was the concentration value or its logarithm used?
7) The calibration curve at the Figure 7c is non-linear. Please recalculate it using two concentration ranges (from 101 to 103 and from 103 to 105).
8) Line 355: “the RctC circuit that describes the layer of MBs with bacteria on the sensor surface” – the modifying layer would demonstrate the Rct and C regardless of the presence of bacteria in the electrode content. Please give some comments about it.
Author Response
Dear Editor,
The authors appreciate the time and effort that reviewers and editor have dedicated to evaluating our manuscript entitled A Cost-Effective and Rapid Manufacturing Approach for Electrochemical Transducers with Magnetic Beads for Biosensing. We believe the reviewers’ insightful comments have greatly improved the clarity and overall quality of the manuscript. In response to the reviewers’ suggestions, we have made the necessary modifications to the manuscript to address the questions and concerns raised. Detailed responses to each comment are provided below, and the corresponding revisions have been highlighted in the revised manuscript.
We hope this revised version of our manuscript meets the expectations of both the reviewer and the editor and is deemed suitable for publication in Micromachines.
Answers to Reviewer 1
Comment 1: The manuscript describes the development of the approach for the rapid fabrication of electrodes with gold and PVC followed by laser ablation. This approach allowed to produce electrodes with different geometry, shape and area. The electrodes were thoroughly tested with a range of electrochemical, microscopic and profile imaging methods. The approach presented is interesting and perspective. Two types of the gold electrodes were used for the fabrication of magnetic beads labeled biosensors for food pathogen detection. To date, devices for food safety control are highly demanded. The references are relevant to the manuscript topic and their quantity is sufficient. This manuscript can be published in Micromachines journal after several minor revisions.
Answer to comment 1: Thank you for your positive evaluation and constructive feedback. We will carefully address the minor revisions to improve the manuscript and ensure it meets the journal's standards.
Comment 2: What was the pH value of PBS used?
Answer to comment 2: The pH value of the PBS used in our experiments was 7.4.
Comment 3: Line 247: HMDS - add the abbreviation description to the manuscript text.
Answer to comment 3: Thank the reviewer for the suggestion. We will include the full description of the HMDS abbreviation in the manuscript text - Hexamethyldisilane (HMDS).
Comment 4: Lines 28, 151: replace “labeling “with “labeled”, Lines 271, 292-293: replace “anionic” and “cationic” with “anodic” and “cathodic”, Lines 353-354: replace “electronic” with “electric”, Line 220: replace “spiked” with “were spiked”.
Answer to comment 4: Thank you for the careful review and valuable suggestions. We will incorporate the recommended corrections to improve the clarity of the manuscript.
Comment 5: There is a poor phrase in the manuscript: Line 441: “The different electrode geometries, including rhomboidal GLE and interdigitated configurations, were shown demonstrating excellent stability, reproducibility, and sensitivity, with minimal signal variation across series of electrodes.”
Answer to comment 5: Thank you for your feedback. We will rephrase the sentence to improve clarity and readability while maintaining its intended meaning.
The various electrode geometries, including rhomboidal GLE and interdigitated configurations, exhibited excellent stability, reproducibility, and sensitivity, with minimal signal variation across multiple electrode series.
Comment 6: Figures 4e and 4f are the same. Please insert the appropriate Figure.
Answer to comment 6: Thank you for noticing this mistake. We will replace the duplicate figure with the correct one in the revised manuscript.
Comment 7: Please check the captions to Figures 6c and 7c. Was the concentration value or its logarithm used?
Answer to comment 7: Thank you for your observation. We will carefully review the captions for Figures 6c and 7c and make necessary corrections.
Comment 8: The calibration curve at the Figure 7c is non-linear. Please recalculate it using two concentration ranges (from 101 to 103 and from 103 to 105).
Answer to comment 8: Thank you for your suggestion.
It should be noted that after 4 CFU/mL, the sensor slowly enters saturation, therefore, the last point has the largest deviation from the linear curve. If we exclude the last point, R2 is 0.9424, demonstrating satisfactory linearity, which is further commented on in the paper L: 393-396. The linearity is also shown below in accordance with the reviewer's request. In the revised manuscript, we have included both curves, linear and linear divided into two concentration ranges.
Comment 9: Line 355: “the RctC circuit that describes the layer of MBs with bacteria on the sensor surface” – the modifying layer would demonstrate the Rct and C regardless of the presence of bacteria in the electrode content. Please give some comments about it.
Answer to comment 9: We provide additional clarification, and this part has been reformulated in the paper to be more accurate, L:362-365. RctC represents the resistance to charge transfer at the electrode/electrolyte interface. The major change in the equivalent Randles circuit was changes in the charge transfer resistance Rct, that is related to the mass transfer phenomenon and/or the dielectric or conductive properties of the captured MBs with bacterial cells. In our biosensors, Rct changes when the MBs or MBs with bacteria are present at the electrode surface, affecting the electron transfer kinetics. A decrease in Rct is visible when just MBs are present on the sensor surface due to improved surface conductivity, while the increase in Rct indicates change due to pathogen binding.

Reviewer 2 Report
Comments and Suggestions for Authors
The presented MS 'A Cost-Effective and Rapid Manufacturing Approach for Electrochemical Transducers with Magnetic Beads for Biosensing' is very interesting and provide a new approach for fabricating the gold electrodes for electrochemical sensing.
The MS also details and explain the detection of bacteria with an efficient way with high sensitivity.
However, i have some concerns that reference electrode is not with Ag/AgCl, how much impact it has in the sensitivity of the electrochemical detection.
it would be good if author can provide a comparison table for detection of similar sample types to present the detection sensitivity in their method.
the similar masking is also used in sputtering system to create the gold electrode, how this approach is efficient way? it is also wasting the cutting of gold material after laser cutting?
Author Response
Dear Editor,
The authors appreciate the time and effort that reviewers and editor have dedicated to evaluating our manuscript entitled A Cost-Effective and Rapid Manufacturing Approach for Electrochemical Transducers with Magnetic Beads for Biosensing. We believe the reviewers’ insightful comments have greatly improved the clarity and overall quality of the manuscript. In response to the reviewers’ suggestions, we have made the necessary modifications to the manuscript to address the questions and concerns raised. Detailed responses to each comment are provided below, and the corresponding revisions have been highlighted in the revised manuscript.
We hope this revised version of our manuscript meets the expectations of both the reviewer and the editor and is deemed suitable for publication in Micromachines.
Answers to Reviewer 2
Comment 1: The presented MS 'A Cost-Effective and Rapid Manufacturing Approach for Electrochemical Transducers with Magnetic Beads for Biosensing' is very interesting and provide a new approach for fabricating the gold electrodes for electrochemical sensing.
The MS also details and explain the detection of bacteria with an efficient way with high sensitivity.
Answer to comment 1: Thank you for your positive feedback and appreciation of our work. We are glad you find our approach innovative and will ensure the manuscript is further refined for clarity and accuracy.
Comment 2: However, i have some concerns that reference electrode is not with Ag/AgCl, how much impact it has in the sensitivity of the electrochemical detection.
Answer to comment 2: Thank you for your thoughtful comment. In planar three-electrode electrochemical sensors, the reference electrode plays a crucial role in providing a stable reference potential. While Ag/AgCl is commonly used due to its well-defined and stable reference potential, using a gold reference electrode may lead to just slight variations in the measured potential. However, the gold reference electrode can still provide reliable performance depending on the specific electrochemical system and measurement methods used. The stability of the signal through several CV cycles in our experiments confirms its stability. In addition, EIS measurements were performed in a two-electrode system, so this does not affect biosensors performances.
Comment 3: it would be good if author can provide a comparison table for detection of similar sample types to present the detection sensitivity in their method.
Answer to comment 3: Thank you for your valuable suggestion. We understand the importance of providing a comparison table, and we will include one in the revised manuscript. However, we would like to emphasize that the primary focus of this work is to demonstrate a novel, cost-effective technology for electrode fabrication, which offers a promising alternative to expensive methods. Our approach can achieve the same detection sensitivity and limit of detection for pathogen detection without the need for additional nanomaterials for signal enhancement. The comparison table will highlight that our method provides excellent performance , making it a competitive low-cost option for biosensing applications. Table 1 was added showing the comparison of the proposed biosensors with similar previously proposed biosensors reported in literature used to determine concentration of different pathogens. In addition, further explanation was added in the discussion section L448-484, and additional references [36]-[42] were added.
Table 1. Comparison of the proposed gold leaf MBs-labeled biosensors with previously reported biosensors reported in literature.
Biosensor type |
Target |
Detection Range (CFU/mL) |
LoD (CFU/mL) |
Reference |
Electrochemical |
S. Typhimurium |
101 - 105 |
10 |
[36] |
Microfluidic |
S. Typhimurium |
102 - 107 |
43 |
[37] |
Electrochemical |
E. coli |
266 - 106 |
80 |
[38] |
Surface-Enhanced Raman Scattering |
L. monocytogenes |
102 - 106 |
12 |
[39] |
Electrochemical |
L. monocytogenes |
101 - 104 |
8.4 |
[40] |
Electrochemical |
L. monocytogenes |
101 - 109 |
2 |
[41] |
Electrochemical |
L. monocytogenes |
103 - 108 |
103 |
[42] |
This work |
S. Typhimurium L. monocytogenes |
101 - 105 |
1.74 2.65 |
|
Comment 4: the similar masking is also used in sputtering system to create the gold electrode, how this approach is efficient way? it is also wasting the cutting of gold material after laser cutting?
Answer to comment 4: We would like to thank the reviewer for their insightful comment. While sputtering systems create gold electrodes with flat films with layer thicknesses in the nanometer range, our laser-based technique results in high surface roughness on the scale of tens of micrometers, which significantly enhances the sensor’s sensitivity. This roughness creates a higher surface area and improves the interaction between the electrode and the target molecules, achieving similar effects to electrodes functionalized with nanoparticles or nanomaterials. In addition, our technology offers several advantages in terms of efficiency, simplicity of manufacturing, and cost-effectiveness. Unlike sputtering, which requires clean room facilities, mask preparation, and expensive equipment, our method does not rely on such complex fabrication steps and expensive equipment. We recognized that material wastage is inherent in our approach, as well as in other techniques such as sputtering or photolithography. However, our method provides a more cost-effective and accessible alternative since it does not require additional chemicals, masks, or specialized equipment. Additional clarification was added in L:433-435 of the revised manuscript.
